# ECMO with vasopressor use during early endotoxic shock: Can it improve circulatory support and regional microcirculatory blood flow?

**Thornton S. Mu** [1]☯*, **Amy M. Becker** [2]☯, **Aaron J. Clark** [2], **Sherreen G. Batts** [2], **Lee-Ann M. Murata** [3], **Catherine F. T. Uyehara** [3]

1 Department of Pediatrics, Brooke Army Medical Center, San Antonio, Texas, United States of America,
2 Department of Pediatrics, Tripler Army Medical Center, Honolulu, HI, United States of America,
3 Department of Clinical Investigation, Tripler Army Medical Center, Honolulu, HI, United States of America

☯ These authors contributed equally to this work.
* thornton.s.mu.mil@mail.mil

## Abstract

### Introduction

While extracorporeal membrane oxygenation (ECMO) is effective in preventing further hypoxemia and maintains blood flow in endotoxin-induced shock, ECMO alone does not reverse the hypotension. In this study, we tested whether concurrent vasopressor use with ECMO would provide increased circulatory support and blood flow, and characterized regional blood flow distribution to vital organs.

### Methods

Endotoxic shock was induced in piglets to achieve a 30% decrease in mean arterial pressure (MAP). Measurements of untreated pigs were compared to pigs treated with ECMO alone or ECMO and vasopressors.

### Results

ECMO provided cardiac support during vasodilatory endotoxic shock and improved oxygen delivery, but vasopressor therapy was required to return MAP to normotensive levels. Increased blood pressure with vasopressors did not alter oxygen consumption or extraction compared to ECMO alone. Regional microcirculatory blood flow (RBF) to the brain, kidney, and liver were maintained or increased during ECMO with and without vasopressors.

### Conclusion

ECMO support and concurrent vasopressor use improve regional blood flow and oxygen delivery even in the absence of full blood pressure restoration. Vasopressor-induced selective distribution of blood flow to vital organs is retained when vasopressors are administered with ECMO.

**Data Availability Statement:** All relevant data are within the paper.

**Funding:** Funding for this research was provided by the Defense Medical Research and Development Program grant #D10_I_AR_J6_925 to PI Catherine FT Uyehara. The funder did not play any role in study design, data collection/analysis, manuscript preparation, or decision to publish.

**Competing interests:** The authors have declared that no competing interests exist.

## Introduction

Shock is defined as a state of oxygen delivery that is inadequate to meet vital organ metabolic demands. [1] Uncorrected hypoperfusion of vital organs leads to cell injury, multiple organ dysfunction syndrome, and death [2]. More specifically, septic (endotoxin-mediated) shock is characterized by an initial peripheral vascular vasodilation with compensatory systemic vaso-constriction as a result of endogenous cardiovascular hormone release in response to hypoten-sion. While preferential regional microcirculatory blood flow (RBF) to preserve vital organ perfusion is essential for survival, maintaining global perfusion is also important for avoidance of long term morbidity and complications of multi-system organ failure. Understanding the effects of shock on whole body and RBF, oxygen consumption, and oxygen delivery are key to guiding resuscitation efforts and improving outcomes. Thus, the therapeutic goal for endotoxic (ET) shock is to achieve hemodynamic stability and to restore microcirculatory flow to all organs [3–5].

Extracorporeal Membrane Oxygenation (ECMO) has been used in patients with pulmo-nary hypertension, respiratory failure, sepsis, and cardiac failure. In children with refractory septic shock, ECMO has been able to successfully support circulation until antibiotics and additional therapies are effective [6–9]. While ECMO has been used in patients with sepsis, its effect on regional microcirculatory blood flow distribution has not fully been characterized. In one study of healthy animals, no difference in regional microcirculatory blood flow was noted on ECMO [10], but it is unclear whether ECMO maintains adequate distribution of RBF to essential organs in a physiologically stressed state. Further, it is unclear whether matching ECMO flow to restore appropriate MAP during shock is the most appropriate goal. Recent adult studies have questioned the emphasis on blood pressure restoration and vasopressor use in hemorrhagic shock versus a permissive hypotension approach [1,11–13] and similar alarms regarding its effects in septic shock and children [14–16].

In this current study, we used an ET shock model to determine if concurrent vasopressor use with ECMO provides necessary circulatory support to maintain oxygen delivery and redis-tribution of blood flow to vital organs.

## Materials and methods

This study was approved by the Institutional Animal Care and Use Committee at Tripler Army Medical Center. Investigators complied with the policies as prescribed in the National Research Council's "Guide for the Care and Use of Laboratory Animals" and the USDA Ani-mal Welfare Act. Animals were handled in accordance with the National Institute of Health (NIH) guidelines in facilities fully accredited by the American Association for Accreditation of Laboratory Animal Care International.

### Animal preparation

Yorkshire cross piglets (Oshiro Farms, Waianae, HI) weighing 7–9 kg were sedated before the induction of anesthesia with the following regimens: acepromazine 1.1 mg/kg IM plus Keta-mine 22–33 mg/kg IM plus atropine 0.02–0.25 mg/kg IM or with a sedation dose of pentobar-bital 30–50 mg/kg intraperitoneal. Anesthesia was maintained with pentobarbital as the anesthetic agent and pigs never regained consciousness throughout the experiment. Pigs were mechanically ventilated to maintain constant tidal volume and end-tidal $CO_2$. To assess physi-ologic measurements, the following catheters were inserted: a 5–6 Fr Swan Ganz catheter with thermistor positioned in the abdominal aorta via left femoral artery access for arterial blood sampling and blood pressure monitoring. A 5–6 Fr Swan Ganz catheter was positioned in the pulmonary artery via left femoral vein access for monitoring of pulmonary pressure,

pulmonary artery wedge pressure, and central venous blood sampling. Hemodynamic measurements were recorded continuously throughout the experiment. Cardiac output (CO) was measured via thermodilution of cold 5% dextrose solution injection into Swan Ganz proximal port in right atrium and temperature change measured by a thermistor in main pulmonary artery. When on ECMO, cold bolus was delivered into the left ventricle and temperature change was measured by thermistor in abdominal aorta; the CO measurements thus reflect the total CO of the contributions of both the native heart output flow and ECMO flow. A left ventricle catheter (3.5 Fr) was placed via left internal carotid artery for left ventricular pressure monitoring and injection of colored microspheres for regional blood flow determination. Blood samples were obtained for blood gases and co-oximetry to determine oxygen utilization. Placement of all catheters was verified by observance of characteristic pressure wave forms. A constant infusion of normal saline at 0.1 ml/kg/min i.v. was maintained throughout the experiment to maintain hydration. A Foley catheter was sutured in the bladder through a lower midline mini-laparotomy to monitor urine output throughout the experiment.

## Extracorporeal circuit

ECMO cannulae were placed for veno-arterial (VA) ECMO in the ET-ECMO groups with a 12–14 Fr cannula aligned with the right atrium via right external jugular vein access, 8–10 Fr cephalad cannula via right external jugular vein (to provide additional venous return), and 8–10 Fr arterial cannula placed at the aortic arch via left carotid artery. ECMO circuits were primed via $CO_2$ introduction, 0.9% saline solution, and filled with whole blood harvested within 24hrs from donor pigs (Oshiro Farms, Waianae, HI). Heparin, sodium bicarbonate, and calcium chloride (at one hundred units each) were added to the circuit. The total circuit priming volume was 450 ml. ECMO flow was achieved via centrifugal pump (Jostra-Rotaflow HL20, Maquet, Rastatt, Germany) through a hollow fiber membrane oxygenator (Quadrox, Josta, Maquet, Rastatt, Germany). All piglets received an initial heparin bolus about 75–100 units/kg before cannulation. Activated clotting time (ACT) samples were drawn periodically and additional heparin boluses were given to maintain ACT between 150 and 200 seconds. Once on ECMO, flow was gradually increased to maintain a target 80-100ml/kg/min goal with oxygenation set at 21% $FiO_2$ and sweep gas rates (300–400 ml/min) adjusted to maintain $pCO_2$ at 35–45 mmHg. Conventional ventilatory support was adjusted for lung rest strategy (respiratory rate halved, baseline positive end expiratory pressure increased, inspiratory time increased, 21% $FiO_2$).

## Experimental protocol

After catheters were placed, a 60 to 120 minute baseline period was allowed (Fig 1). After baseline measurements were obtained, piglets were randomized to one of 4 groups. The Control group (n = 6), of animals were prepared with all ECMO catheter placements, but not put on ECMO, and monitored continuously for the duration of the experiment with additional measurements obtained at 90 minutes and 3 hours after baseline. Group 2 received ET with no treatment (n = 7), group 3 received ET with ECMO (n = 6), and group 4 received ET with ECMO plus vasopressor treatment (n = 8). For groups 2, 3, and 4, ET shock was induced in piglets with i.v. boluses of *E. coli* endotoxin (lipopolysaccharide purified from Escherichia coli serotype 055:B55, cat. No L-2637, 5,000 to 35,000 units, Sigma, St Louis, MO) in 5,000 U increments until a steady state of hypotension was achieved, defined as a minimum 30% decrease in MAP. Initial endotoxin shock state measurements were obtained at 1 hour of hypotension. Repeat measurements for animals in group 2 were obtained after 3 additional hours of hypotension. Group 3 animals were placed on VA ECMO after the initial 1 hour hypotensive period

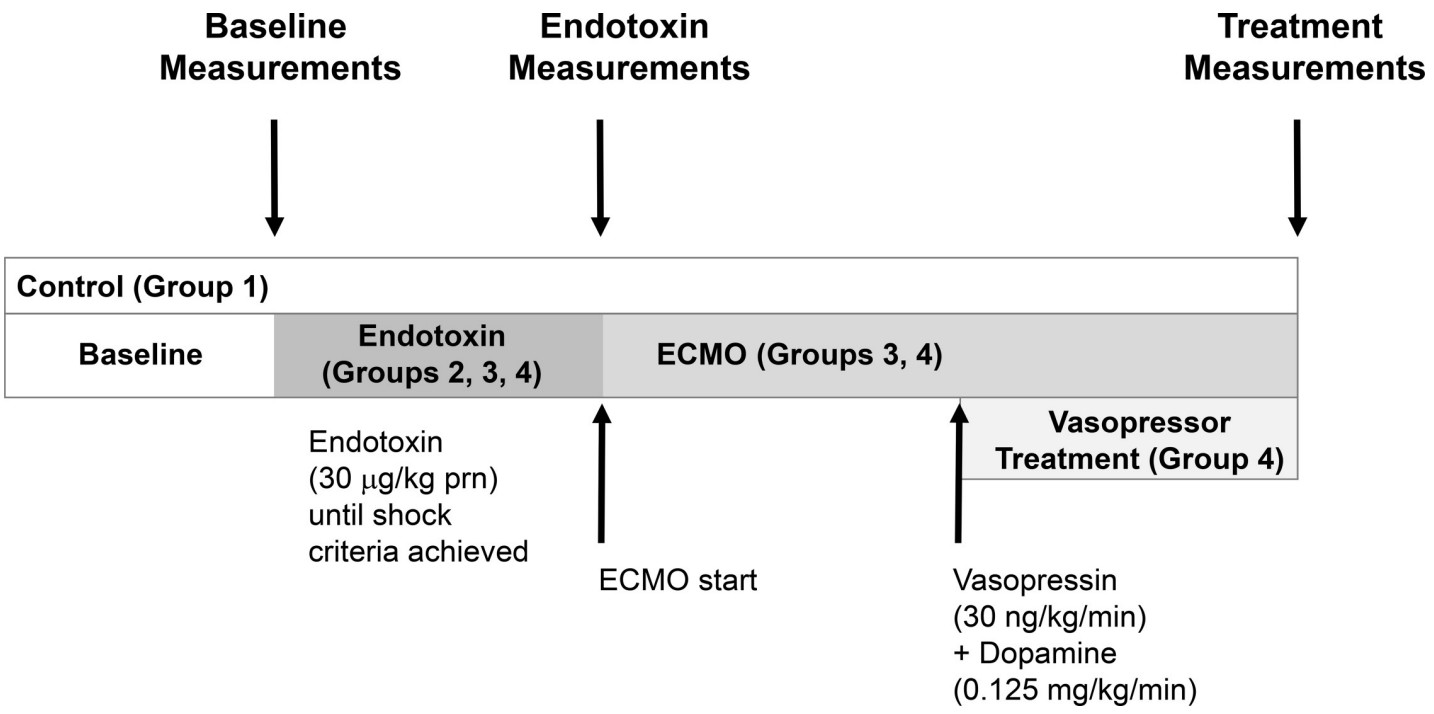

**Fig 1. Protocol timeline.**

and measurements were obtained after animals had been on ECMO for 3 hours. Animals in group 4 were placed on VA ECMO after the initial 1 hour period of hypotension. After 1 hour on VA ECMO, they received a combination pressor treatment of dopamine (dose titrated 0.006–0.125 mg/kg/min) and vasopressin (dose constant at 30 ng/kg/min) infusions for 2 hours to achieve the goal of returning MAP back to baseline levels.

### Assessment of Regional Microcirculatory Blood Flow (RBF)

During each sampling period, RBF was assessed via injection of colored microspheres ($2 \times 10^6$ microspheres) over 20 seconds into the left ventricular catheter, followed by saline flush over 10 seconds. Blood was collected at 2ml/min over 70 seconds through the left femoral artery beginning 10 seconds before microsphere injection for assessment of cardiac output at the time of microsphere injection and calculation of RBF based on microsphere concentration (ml/min per gram of tissue). A different colored microsphere was used for each measurement period to distinguish RBF at different periods.

All animals were euthanized with Beuthanasia-D (phenytoin/pentobarbital) and tissues harvested for microsphere extraction. Tissue samples were obtained from skin, muscle, cerebrum, cerebellum, midbrain, brainstem, kidney, heart, liver, small intestine, large intestine, and stomach.

### Statistical analyses

Two-way Analysis of variance (ANOVA) with repeated measures over time was used to compare hemodynamic measurements, RBF, and oxygen utilization values at the three measurement periods (baseline, after established endotoxin-induced shock, and following treatment within groups) between and within groups (JMP 4.0.4 program, SAS Institute, Cary, NC). Group by period interaction effects that were significant in the overall ANOVA were

compared with post hoc orthogonal contrast tests to discern differences between baseline versus ET shock state and ET shock state versus each treatment. Data are expressed as mean ± standard error of the mean. A p value of <0.05 was considered to be significant.

## Results and discussion

**Hemodynamics.** All control group hemodynamic parameters remained constant at all time points. Table 1 Endotoxin caused a dramatic decrease in MAP from baseline due to a drop in CO without a compensatory increase in systemic vascular resistance (SVR) sufficient to prevent hypotension. ECMO support alone, delivered to achieve baseline CO, did not return MAP to baseline. The MAP of untreated ET (group 2) and ET-ECMO (group 3) remained decreased at 61±5 and 63±9 mmHg, respectively. Treatment with vasopressin and dopamine in addition to ECMO (group 4) increased SVR above baseline levels. Vasoconstriction was not changed with ECMO alone, but vasopressors helped increase perfusion pressure with increased SVR. In group 4, vasopressor support returned MAP to baseline levels. ET caused CO to decrease in groups 2, 3, and 4 (236 ± 10 to 145 ± 9ml/kg/min (p<0.05)). ECMO helped improve CO in group 3. A compensatory increase in HR in response to shock occurred with ET, but with improved CO on ECMO, the tachycardia resolved and returned to baseline.

During endotoxic shock, decreased perfusion was associated with decreased mixed venous oxygen saturations, indicative of increased oxygen extraction at the tissue level. ECMO with or without vasopressors increased oxygen delivery, and hence was able to improve SvO2 to pre-shock levels.

ET induced pulmonary hypertension in groups 2, 3, and 4, which persisted in non-ECMO treated animals in group 2. ECMO increased oxygen delivery and thus ameliorated hypoxemia

**Table 1. Hemodynamics.**

|  |  | Control | ET | ET+ECMO | ET+ECMO+Pressors |
|---|---|---|---|---|---|
| **MAP (mm Hg)** | Baseline | 95±3 |  | 90±2 |  |
|  | Endotoxin | 93±4 |  | 55±2[a] |  |
|  | Treatment | 89±3 | 61±5[a] | 63±9[a] | 100±5[bc] |
| **CO (ml/min/kg)** | Baseline | 193±17 |  | 236±10 |  |
|  | Endotoxin | 178±17 |  | 145±9[a] |  |
|  | Treatment | 171±12 | 132±17[a] | 178±21[ab] | 118±8[abc] |
| **SVR (mmHg/(L/min/kg)** | Baseline | 459±21 |  | 387±20 |  |
|  | Endotoxin | 532±44 |  | 387±32 |  |
|  | Treatment | 522±39 | 447±32 | 329±25 | 814±48[abc] |
| **SV (ml/beat)** | Baseline | 1.2±0.1 |  | 1.6±0.1 |  |
|  | Endotoxin | 1.1±0.1 |  | 0.7±0.1[a] |  |
|  | Treatment | 0.9±0.1 | 0.7±0.2[a] | 1.2±0.2[a] | 0.8±0.1[ab] |
| **HR (beats/min)** | Baseline | 163±13 |  | 155±5 |  |
|  | Endotoxin | 169±13 |  | 207±8[a] |  |
|  | Treatment | 186±8 | 200±18[a] | 163±17[b] | 147±12[b] |
| **SvO2** | Baseline | 0.71±0.02 |  | 0.78±0.02 |  |
|  | Endotoxin | 0.71±0.02 |  | 0.54±0.03[a] |  |
|  | Treatment | 0.72±0.01 | 0.47±0.05[a] | 0.84±0.03[b] | 0.78±0.04[b] |

Values represent mean ± s.e.m.

[a] = significantly different from baseline, p<0.05.

[b] = significantly different from ET during endotoxin period 2, p<0.05.

[c] = significant difference between ECMO only and ECMO plus vasopressors, p<0.05

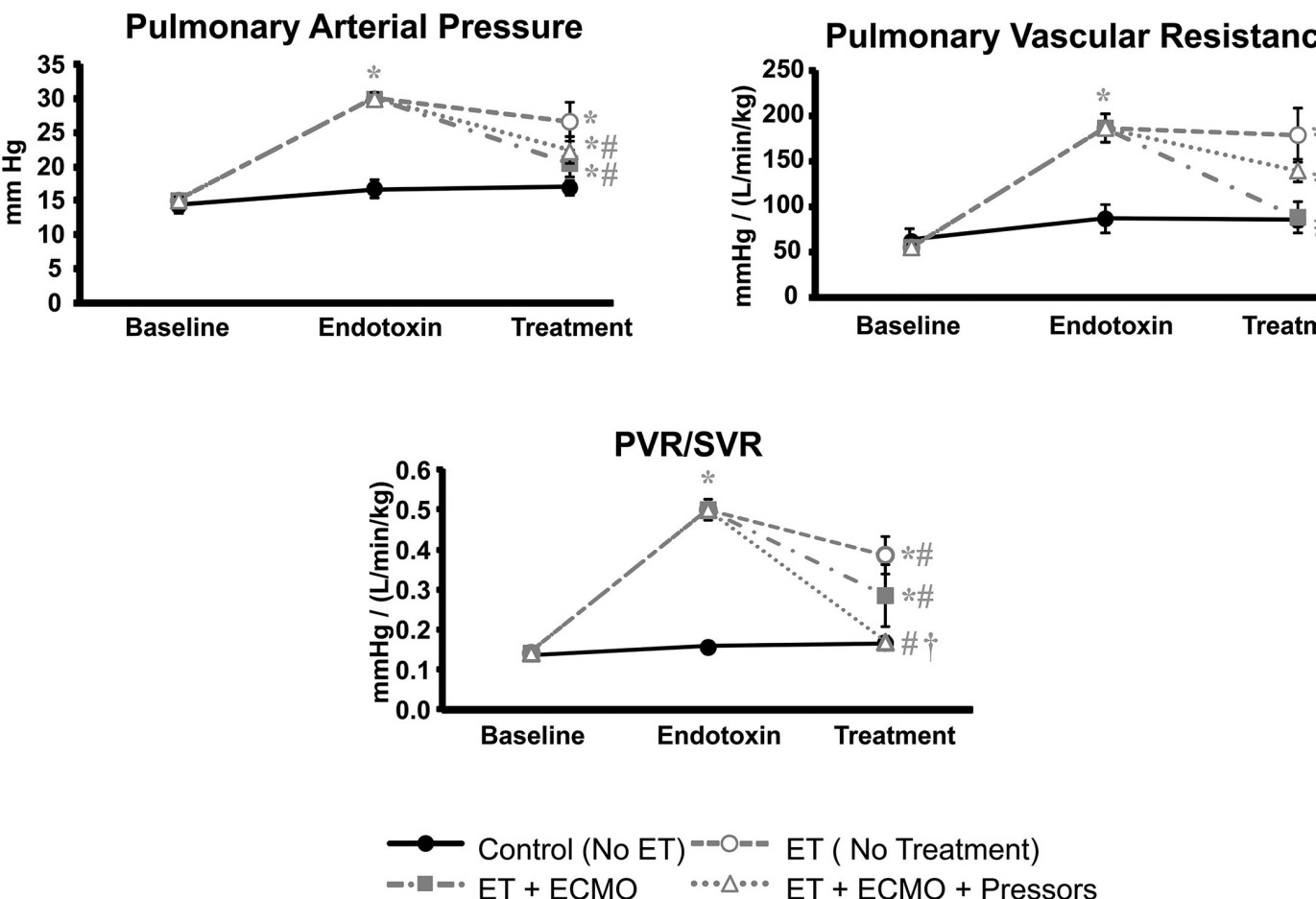

**Fig 2. Pulmonary function.** A. Pulmonary Artery Pressure B. Pulmonary Vascular Resistance C. PVR/SVR ratio. Values represent mean ± s.e.m. * = significantly different from baseline, p<0.05. # = significantly different from ET during endotoxin period 2, p<0.05. † = significant difference between ECMO only and ECMO plus vasopressors, p<0.05.

which led to some improvement in pulmonary pressure (Fig 2A). With improved oxygenation, ECMO decreased PVR to baseline levels in groups 3 and group 4 (Fig 2B). The PVR/SVR ratio was normalized in group 4 with the addition of vasopressors (Fig 2C).

### Oxygen utilization

ET caused a decrease in overall oxygen delivery (Fig 3A). This decreased oxygen delivery continued in untreated ET shock animals. In animals placed on ECMO, O2 delivery improved towards baseline and was significantly higher than untreated ET, but did not return to baseline due to persistently decreased cardiac output. The addition of vasopressors did not further improve overall oxygen delivery.

Despite the differences in decreased delivery, oxygen consumption was sustained in ET animals through increased oxygen extraction (Fig 3B and 3C). When animals were placed on ECMO, oxygen consumption and extraction were decreased from the ET period, likely due to the restoration of blood flow with VA ECMO. Vasopressin and dopamine pressor treatment did not alter the effect of ECMO on oxygen consumption and extraction. As ECMO improved systemic circulation and decreased tachycardia, the decreased oxygen consumption is consistent with a reduced cardiac work with ECMO.

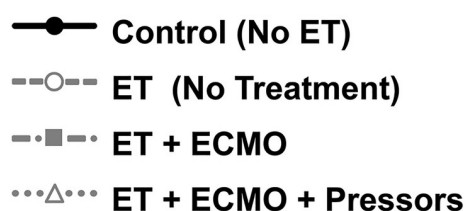

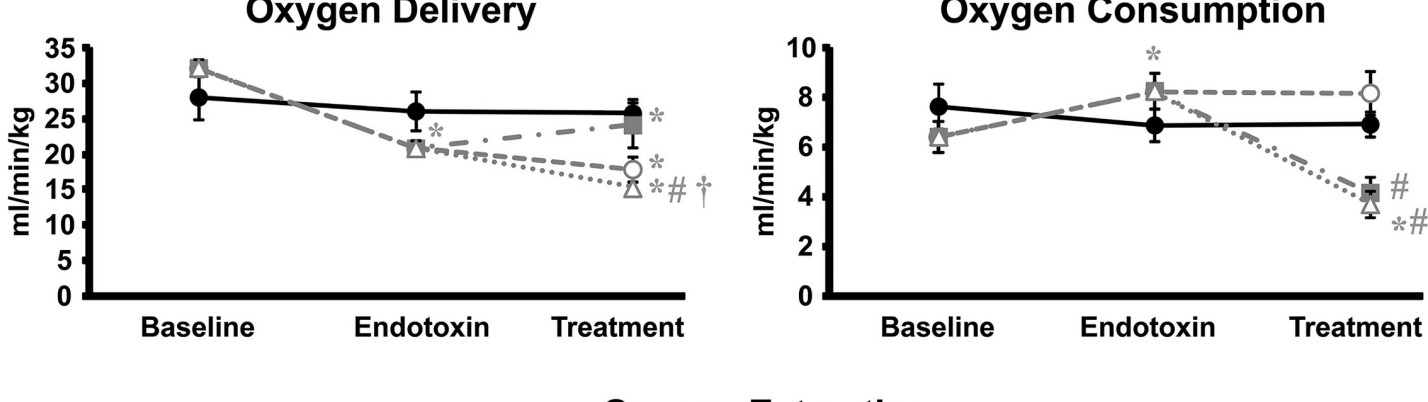

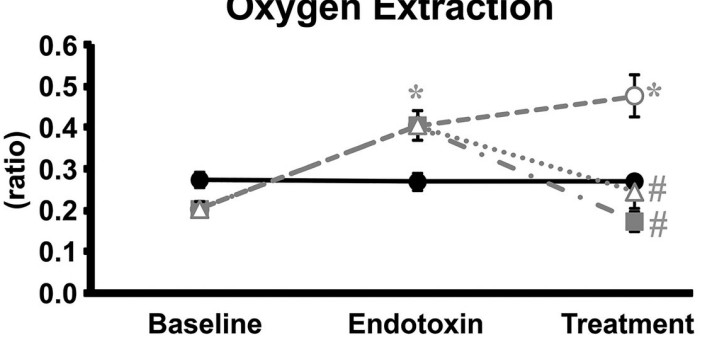

**Fig 3. Oxygen utilization.** A. Oxygen delivery B. Oxygen consumption C. Oxygen extraction. Values represent mean ± s.e.m. * = significantly different from baseline, $p < 0.05$. # = significantly different from ET during endotoxin period 2, $p < 0.05$. † = significant difference between ECMO only and ECMO plus vasopressors, $p < 0.05$.

### Regional microcirculatory blood flow

For the various organs examined, RBF remained unchanged at all time periods in the control group. Table 2 RBF to all areas of the brain was maintained in all groups, showing preservation of cerebral blood flow in early septic shock. Blood flow to different regions of the brain (brainstem, midbrain) increased slightly in the endotoxin groups.

Blood flow was maintained to the heart with ET despite the decrease in MAP. ECMO with and without pressors decreased heart microperfusion below ET levels, likely due to the partial cardiac bypass of flow to the ECMO circuit.

ET caused a decrease in renal blood flow, commensurate with the overall decrease in perfusion pressure with shock. ECMO helped increase overall perfusion and accordingly, renal blood flow. Vasopressors did not alter this ECMO-induced return of renal blood flow towards baseline.

The overall decrease in MAP with ET was also associated with a 60% decrease in liver blood flow. Liver RBF remained depressed with ECMO whereas ECMO with vasopressors helped return liver blood flow towards baseline.

**Table 2. Regional microcirculatory blood flow.**

| | | Control | ET | ET+ECMO | ET+ECMO+Pressors |
|---|---|---|---|---|---|
| **Cerebrum RBF (ml/min/kg)** | Baseline | 0.51±0.12 | | 0.50±0.04 | |
| | Endotoxin | 0.57±0.17 | | 0.52±0.06 | |
| | Treatment | 0.56±0.07 | 0.53±0.15 | 0.53±0.08 | 0.54±0.08 |
| **Cerebellum RBF (ml/min/kg)** | Baseline | 0.57±0.10 | | 0.67±0.04 | |
| | Endotoxin | 0.67±0.16 | | 0.80±0.09 | |
| | Treatment | 0.69±0.06 | 0.73±0.18 | 0.70±0.11 | 0.99±0.16[a] |
| **Midbrain RBF (ml/min/kg)** | Baseline | 0.36±0.05 | | 0.47±0.03 | |
| | Endotoxin | 0.41±0.07 | | 0.57±0.05[a] | |
| | Treatment | 0.44±0.05 | 0.50±0.12 | 0.45±0.06 | 0.65±0.08[a] |
| **Brainstem RBF (ml/min/kg)** | Baseline | 0.36±0.06 | | 0.51±0.03 | |
| | Endotoxin | 0.42±0.08 | | 0.70±0.09[a] | |
| | Treatment | 0.49±0.07 | 0.62±0.16 | 0.48±0.08 | 0.70±0.10 |
| **Heart RBF (ml/min/kg)** | Baseline | 1.71±0.28 | | 1.67±0.12 | |
| | Endotoxin | 1.69±0.38 | | 2.01±0.20 | |
| | Treatment | 1.59±0.22 | 2.31±0.43 | 1.28±0.23[b] | 1.17±0.14[b] |
| **Kidney RBF (ml/min/kg)** | Baseline | 3.00±0.42 | | 2.80±0.14 | |
| | Endotoxin | 2.93±0.36 | | 1.74±0.20[a] | |
| | Treatment | 2.74±0.25 | 1.78±0.32 | 3.00±0.74[b] | 2.57±0.65 |
| **Liver RBF (ml/min/kg)** | Baseline | 0.93±0.22 | | 0.96±0.08 | |
| | Endotoxin | 0.78±0.21 | | 0.33±0.06[a] | |
| | Treatment | 0.51±0.11[a] | 0.28±0.07[a] | 0.21±0.05[a] | 0.64±0.12[abc] |
| **Stomach RBF (ml/min/kg)** | Baseline | 0.23±0.04 | | 0.36±0.03 | |
| | Endotoxin | 0.26±0.08 | | 0.32±0.03 | |
| | Treatment | 0.26±0.04 | 0.37±0.11 | 0.41±0.09 | 0.19±0.02[ac] |
| **Sm Intestine RBF (ml/min/kg)** | Baseline | 0.78±0.15 | | 0.99±0.11 | |
| | Endotoxin | 0.91±0.33 | | 0.68±0.07[a] | |
| | Treatment | 0.80±0.13 | 0.73±0.21 | 1.12±0.25[b] | 0.66±0.06 |
| **Lg Intestine RBF (ml/min/kg)** | Baseline | 0.62±0.08 | | 0.76±0.07 | |
| | Endotoxin | 0.64±0.12 | | 0.51±0.06[a] | |
| | Treatment | 0.82±0.18 | 0.51±0.08 | 0.74±0.16 | 0.55±0.11 |

Values represent mean ± s.e.m.

[a] = significantly different from baseline, p<0.05.

[b] = significantly different from ET during endotoxin period 2, p<0.05.

[c] = significant difference between ECMO only and ECMO plus vasopressors, p<0.05

The ET-induced drop in MAP caused a decrease in blood flow to the small and large intestines but no change in flow to the stomach. ECMO did not affect the flow to the stomach but increased blood flow to the small intestines so that it was no longer depressed from baseline. The addition of vasopressors negated the ECMO effect on the intestines.

## Discussion

Decreased perfusion pressure observed during ET shock redistributes blood in the peripheral circulation and decreases venous return that can lead to cardiac failure. Cardiac depression is proportional to the severity of sepsis [17,18], and cardiovascular support strategies in septic shock need to address prevention of global circulatory dysfunction. Increasing ECMO flow to achieve pre-hypotension baseline MAP may risk over-perfusing certain vascular beds as we

demonstrated in a previous study [19] which showed that using normotensive MAP as a criteria for setting ECMO flow caused an increase in brain blood flow which could lead to cerebral hemorrhage risk. Conversely, if ECMO flow is not high enough to achieve an adequate perfusion pressure, microcirculatory flow to vital organs may be insufficient.

Mixed venous saturation (SvO2) has been used as a clinical indicator of adequate O2 delivery with a therapeutic goal of SvO2 > 70% [20]. Previous to this study, it has been unclear how reduced overall CO while on ECMO would affect blood flow distribution to various vascular beds. Vasopressor therapy is commonly used to achieve normotensive MAP, yet it is unclear how the pressor-induced vasoconstriction may, in itself, reduce flow to certain vascular beds.

In this study we demonstrated that VA ECMO is able to achieve adequate blood flow, and thereby adequate oxygen delivery even when neither MAP nor CO returned to baseline. Persistent pulmonary hypertension contributes to worsening hypoxemia in endotoxic shock. We previously showed that vasopressin spares the pulmonary vasculature of its potent vasoconstrictive action [21], thus it was not surprising to see that when vasopressin and dopamine were used in conjunction with ECMO that the elevated pulmonary to systemic vascular resistance ratio (PVR/SVR) caused by ET returned to baseline. Because oxygen delivery was already increased with ECMO alone, the relative vasoconstrictive sparing of the pulmonary bed did not contribute additional systemic oxygen delivery gains.

There did not seem to be any immediate adverse systemic vasoconstrictive effect seen with the use of vasopressors. In striving for restoration of pre-illness MAP and CO, there may not be a clear benefit to overall total systemic oxygen utilization, however the benefit of using vasopressors may be the redistribution of microcirculatory flow to vital organs. These findings could support recent discussions about permissive hypotension to simply achieve minimal perfusion pressure, and the benefit of vasopressor use in septic shock [12,14,16].

Our study presents a comprehensive picture of RBF in all major organ systems during early ET shock in an ECMO model incorporating vasopressor use. Earlier studies implicated preferential blood flow toward the upper body with ECMO delivery compared with left ventricular outflow supplying blood to the heart and lower body [22]. During early ET shock when endogenous vasopressin is elevated, regional microcirculatory blood flow is directed to certain organs such as the brain and heart, which supports previously reported findings that vasopressin redistributes microcirculatory flow to vital organs and increases brain blood flow in hypotensive shock via selective vasoconstriction of vascular beds [23]. The vasoconstrictive effects of vasopressin and dopamine remained evident even during ECMO. Thus vasopressor agents may be beneficial not only for restoring blood pressure, but also for re-distributing microcirculatory blood flow to maintain vital organ tissue viability in septic shock.

We note with interest that ET increased RBF to midbrain and brainstem but that ECMO normalized it. Vasopressors with ECMO increased brain RBF possibly via redistribution from the intestines. Cardiac RBF was maintained with ET and as might be expected, decreased with partial cardiac bypass on ECMO, and vasopressors did not further alter heart microcirculatory flow. ET also caused a decrease in renal RBF in accordance with decreased perfusion pressure during hypotension. Smith et al reported distribution of RBF toward the brain and away from heart and kidneys in a lamb model on ECMO and found no significant differences in the spleen, lung, or liver [24]. However, our data report that ECMO both alone and with vasopressors returned renal RBF to baseline levels. Blood flow to intestinal organs (stomach, SI, LI) was unchanged or improved in the ECMO only group, but this effect was negated with the addition of vasopressors. The ET-induced decrease in liver blood flow persisted with ECMO but improved with the addition of vasopressors. This effect is likely due to vasopressin which is known to increase liver blood flow [23].

There are limitations to our study. Prolonged hypotension and alterations in microcirculation in sepsis occur over several days, and ECMO therapy and vasopressor requirements can persist for days to weeks. Therefore, our short experimental timeline does not permit interpretation of ECMO (with and without vasopressors) effects beyond the initial period in ET shock. Vasopressin and dopamine were chosen for this protocol based upon investigators' experience with these vasopressors both in clinical and experimental settings. We acknowledge that many other vasopressors are used to manage septic shock, and our findings limit the ability to extrapolate to other commonly used vasopressors, such as norepinephrine or dobutamine. Lastly, we also acknowledge the inherent limitations of using non-human species in translational research.

## Conclusions

In conclusion, extracorporeal support and concurrent vasopressor use sustained and improved regional blood flow and oxygen delivery in this model of early ET shock even in the absence of fully restoring MAP or CO to baseline. Additionally, ECMO and concurrent vasopressors redistribute RBF preferentially to vital organ systems. Results suggest that the therapeutic goal in using ECMO should be to maximize blood flow and oxygen delivery to vital organs rather than simply trying to restore blood pressure *per se*. Further studies, particularly incorporating longer ECMO periods and different vasopressors are needed to fully characterize the effects of ECMO therapy on microcirculatory flow and oxygen utilization in septic shock.

## Acknowledgments

We would like to acknowledge Wayne Ichimura, Aileen Sato, and Claudia Hernandez for all of the support they provided with helping the experimental protocol run smoothly.

Thank you to Glenn Hashiro and Lisa Coviello who helped develop the endotoxin model and earlier hemodynamics assessment procedures for the PVR/SVR and vasopressin data.

"The view(s) expressed herein are those of the author(s) and do not reflect the official policy or position of Brooke Army Medical Center, Tripler Army Medical Center, the US Army Medical Department, the US Army Office of the Surgeon General, the Department of the Army, or the Department of Defense, or the US Government."

## Author Contributions

**Conceptualization:** Amy M. Becker, Sherreen G. Batts, Catherine F. T. Uyehara.

**Data curation:** Thornton S. Mu, Amy M. Becker, Lee-Ann M. Murata, Catherine F. T. Uyehara.

**Formal analysis:** Thornton S. Mu, Amy M. Becker, Sherreen G. Batts, Lee-Ann M. Murata, Catherine F. T. Uyehara.

**Funding acquisition:** Catherine F. T. Uyehara.

**Investigation:** Thornton S. Mu, Amy M. Becker, Aaron J. Clark, Sherreen G. Batts, Catherine F. T. Uyehara.

**Methodology:** Thornton S. Mu, Amy M. Becker, Sherreen G. Batts, Catherine F. T. Uyehara.

**Project administration:** Amy M. Becker.

**Supervision:** Catherine F. T. Uyehara.

**Writing – original draft:** Thornton S. Mu, Amy M. Becker, Catherine F. T. Uyehara.

**Writing – review & editing:** Thornton S. Mu, Amy M. Becker, Aaron J. Clark, Sherreen G. Batts, Lee-Ann M. Murata, Catherine F. T. Uyehara.

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
