## [Decision Letter · Decision Letter 0]

1 Aug 2019

PONE-D-19-20218

ECMO with vasopressor use during early endotoxic shock: Can it improve circulatory support and regional microcirculatory blood flow?

PLOS ONE

Dear Mu,

Thank you for submitting your manuscript to PLOS ONE. After careful consideration, we feel that it has merit but does not fully meet PLOS ONE’s publication criteria as it currently stands. Therefore, we invite you to submit a revised version of the manuscript that addresses the points raised mainly by reviewer 2.

We would appreciate receiving your revised manuscript by Sep 15 2019 11:59PM. To enhance the reproducibility of your results, we recommend that if applicable you deposit your laboratory protocols in protocols.io, where a protocol can be assigned its own identifier (DOI) such that it can be cited independently in the future. For instructions see: http://journals.plos.org/plosone/s/submission-guidelines#loc-laboratory-protocols

We look forward to receiving your revised manuscript.

Kind regards,

Michael Bader

Academic Editor

PLOS ONE

Journal Requirements:

2. At this time, we request that you  please report additional details in your Methods section regarding animal care:

*The method of euthanasia

*The source of the pigs and the source of the donor pig blood used in this study.

* Please clarify whether the pigs were anesthetized throughout the experiments, or whether the ever regained consciousness.

. Thank you for your attention to these requests.

Reviewers' comments:

Reviewer's Responses to Questions

**Comments to the Author**

1. Is the manuscript technically sound, and do the data support the conclusions?

Reviewer #1: Yes

Reviewer #2: Partly

2. Has the statistical analysis been performed appropriately and rigorously? 

Reviewer #1: Yes

Reviewer #2: I Don't Know

3. Have the authors made all data underlying the findings in their manuscript fully available?

Reviewer #1: Yes

Reviewer #2: Yes

4. Is the manuscript presented in an intelligible fashion and written in standard English?

Reviewer #1: Yes

Reviewer #2: Yes

5. Review Comments to the Author

Reviewer #1: Findings are consistent with previous clinical and research studies. A group with shock and vasopressors without echo would have been interesting.

Reviewer #2: Dear authors

It has been a pleasure to write and comment on your article: “ECMO with vasopressor…”

As you can see below, I have something that need changes and I have some suggestions for improvement.

Introduction:

The section is well written and appropriate.

Line 46: MODS as an abbreviation is not necessary because it is not used again in the text.

Materials and Methods:

Even though the local authorities approved the study, the anesthesia was not in accordance with European standard. In Europe, a protocol with Pentobarbital without analgesics cannot be accepted in surgery. Was the anesthesia as described or is anything missing?

How was the ventilator settings during the experiments? FiO2 and how was the ventilator and oxygenator air flow adjusted?

Extracorporeal Circuit:

Please explain why a cephalad 8-10 Fr catheter was inserted in the jugular vein, line 107

On line 108: arterial cannula placed at the left ventricle? Is it in the aortic arch or in the descending aorta at the level of the left ventricle? If placed in the ventricle it may result in valve insufficiency and backward failure to the pulmonary circulation. In line 96 another catheter is placed via the same internal carotid artery and inserted to the left ventricle. I guess this is a smaller catheter, please mention.

A priming volume 450 ml is very big for a 9 kg pig even the circuit was primed with blood. The donor blood can interfere with pulmonary resistance. Did the control group that was not on ECMO receive donor blood?

Why not use an infant set for ECMO, or bigger pigs, instead of using a 50 kg pig as donor.

I think the extra corporeal system was biocoated, if it was possible to run the ECMO with an ACT below 200 seconds.

Experimental protocol:

The first group, if I understand it correct, please indicate that the pigs were prepared for ECMO with all catheters inserted but the ECMO was not started.

Line 125: n=21. Please mention how many in each group first and then about ET treatment.

Why did you end at 27 pigs? How many pigs did you have from the beginning?

Results:

Line 190: be careful not to conclude something of PAP, but only about PVR, because ECMO in itself will bypass blood from the pulmonary circulation.

Line 2019: Maybe better to change CO with systemic circulation

Discussion:

Line 255 to 270 is strictly not discussion but introduction

The biggest limitation is the species difference.

Very best

Benedict Kjaergaard

6. PLOS authors have the option to publish the peer review history of their article (what does this mean?). If published, this will include your full peer review and any attached files.

Reviewer #1: No

Reviewer #2: Yes: Benedict Kjaergaard

---

## [Author Response · Author response to Decision Letter 0]

18 Sep 2019

Hello-

Please see the attached response to reviewers PDF.

---

## [Decision Letter · Decision Letter 1]

25 Sep 2019

ECMO with vasopressor use during early endotoxic shock: Can it improve circulatory support and regional microcirculatory blood flow?

PONE-D-19-20218R1

Dear Dr. Mu,

We are pleased to inform you that your manuscript has been judged scientifically suitable for publication and will be formally accepted for publication once it complies with all outstanding technical requirements.

With kind regards,

Michael Bader

Academic Editor

PLOS ONE

Additional Editor Comments (optional):

Reviewers' comments:

Reviewer's Responses to Questions

**Comments to the Author**

1. If the authors have adequately addressed your comments raised in a previous round of review and you feel that this manuscript is now acceptable for publication, you may indicate that here to bypass the “Comments to the Author” section, enter your conflict of interest statement in the “Confidential to Editor” section, and submit your "Accept" recommendation.

Reviewer #1: All comments have been addressed

Reviewer #2: All comments have been addressed

2. Is the manuscript technically sound, and do the data support the conclusions?

Reviewer #1: (No Response)

Reviewer #2: Yes

3. Has the statistical analysis been performed appropriately and rigorously? 

Reviewer #1: (No Response)

Reviewer #2: Yes

4. Have the authors made all data underlying the findings in their manuscript fully available?

Reviewer #1: (No Response)

Reviewer #2: Yes

5. Is the manuscript presented in an intelligible fashion and written in standard English?

Reviewer #1: (No Response)

Reviewer #2: Yes

6. Review Comments to the Author

Reviewer #1: Revisions complete. Comments worthwhile and appropriately placed in manuscript. Nothing further to add to this review.

Reviewer #2: I find the changes acceptable for publication. Especially that you changed the sections of medication, cannulas and ventilator settings. Your explanation of the statistics is OK.

7. PLOS authors have the option to publish the peer review history of their article (what does this mean?). If published, this will include your full peer review and any attached files.

Reviewer #1: Yes: Irving Kron

Reviewer #2: Yes: Benedict Kjaergaard

---

## [Editor Report · Acceptance letter]

2 Oct 2019

PONE-D-19-20218R1 

ECMO with vasopressor use during early endotoxic shock: Can it improve circulatory support and regional microcirculatory blood flow? 

Dear Dr. Mu:

I am pleased to inform you that your manuscript has been deemed suitable for publication in PLOS ONE. Congratulations! Your manuscript is now with our production department. 

With kind regards,

on behalf of

Prof. Michael Bader 

Academic Editor

PLOS ONE